# 3D DEM Analysis of Particle Breakage Effect on Direct Shear Tests of Coarse Sand

**DOI:** 10.3390/ma16145025

**Published:** 2023-07-16

**Authors:** Mohamed Amine Benmebarek, Majid Movahedi Rad, Sadok Benmebarek

**Affiliations:** 1Department of Structural and Geotechnical Engineering, Faculty of Architecture, Civil Engineering and Transport Sciences, Széchenyi István University, Egyetem tér 1, 9026 Győr, Hungary; benmebarek.mohamed.amine@hallgato.sze.hu; 2NMISSI Laboratory, Department of Civil Engineering and Hydraulic, Biskra University, Biskra 07000, Algeria; s.benmebarek@univ-biskra.dz

**Keywords:** DEM, direct shear, granular material, crushing evolution, contractive behavior

## Abstract

This paper explores the effect of particle breakage on the mechanical behavior of coarse sand through 3D Discrete Element Method (DEM) simulations of direct shear tests (DST). The objective is to gain insights into the macro- and micro-mechanical behaviors of crushable coarse sand, with a particular focus on the stress–strain relationship, volumetric deformation, and evolution of grain crushing. The simulations involve a comparison between non-crushable and crushable particle models, where the crushable particles are implemented in the shear zone of the DST subjected to different high normal stresses. The findings indicate that the crushable particles experience partial crushing at peak shear stress, with further particle crushing leading to the production of finer particles at the shearing plane during shearing at the critical state. The migration of these finer particles under pressure and gravity generates their accumulation predominantly in the lower section of the simulation box. Importantly, the presence of crushing in the DST induces a decrease in the shear stress and an increase in the volumetric strain leading to contractive behavior instead of dilation, which gradually stabilizes the volumetric deformation at higher normal stresses.

## 1. Introduction

From field practice and laboratory tests, it is widely observed that stress levels sufficient for particle breakage are encountered in deep penetration and compaction, and examples of such situations include displacement pile driving, cone penetration testing in coarse-grained soils and high earth or rockfill dams. Particle breakage is a phenomenon that is commonly observed in offshore engineering practices, occurring in various types of soil particles, including both strong particles like quartz sand, as well as weaker ones such as carbonate or calcareous sands [1,2,3,4,5,6,7,8,9,10,11,12,13,14,15,16,17,18,19,20,21,22,23,24,25,26,27].

Almeida et al. [1] noticed that the increase in cone resistance with the relative density of calcareous crushable sand is noticeably different from that of uncrushable silica sand. This difference poses challenges when applying the correlations developed for silica sands to crushable grain materials [2]. Angemeer et al. [3] have found that, for offshore pile load testing in calcareous sands, pile frictional capacities in cohesionless soils may be as low as one-tenth of the values typically used in conventional design. Coop and Airey [4] have highlighted the vulnerability of calcareous sand to crushing, noting that it can occur even at extremely low-stress levels, such as those at or below 1 MPa.

Therefore, the issue of particle breakage has garnered significant attention from researchers within the field of geotechnical engineering, primarily due to the frequent occurrence of crushable materials. Several researchers have conducted calibration chamber and centrifuge tests on carbonate sands to better understand what happens when a pile is driven into weak sands [5,6,7,8]. This phenomenon has also attracted many researchers to conduct laboratory tests to study the influence of particle breakage on the critical state behaviors, shear stress ratios and dilatancy of granular soils [9,10,11,12,13,14,15,16,17,18,19,20,21,22,23,24,25,26,27] as well as numerical modeling [14,28,29,30,31,32,33,34,35,36,37,38,39,40,41,42,43,44,45,46,47,48,49,50,51].

Conducting calibration chamber tests, White and Bolton [6] observed a conical zone of particle crush under the pile tip and an interface zone around the pile shaft comprising thin crushed soil particles contracted while shearing along the shaft interface. Recently, using the approach of analyzing sequential images of the sand under the tip of the pile, taken sequentially from the top with the layer-by-layer removal of the sand via a vacuum cleaner, Mao et al. [7] graphically illustrated the shape of the particle breakage zone under the tip of a flat-ended pile. They found that the breakage zone is not uniformly distributed below the pile tip, but is concentrated in the conically shaped shear zone.

Under high stresses, it is observed that crushing at sliding contacts affects stress–strain behavior and decreases the rate of dilation for both triaxial tests [12,13,14] and direct and ring shear tests [15,16,17,18,19,20,21,22,23,24,25]. In their study, Nakata et al. [12] performed triaxial compression tests on Aio sand and observed that under relatively low confining pressures (specifically, 2.94 MPa, which was lower than the particle strength), the predominant modes of particle breakage were abrasion and asperity breakages. With an increase in the confining pressure, the occurrence of splitting breakage began to appear. Notably, at a high confining pressure of 9.81 MPa, splitting breakage emerged as the dominant mode of particle breakage.

Coop et al. [15] utilized ring shear tests on carbonate specimens to investigate particle crushing behavior. Their findings revealed that the extent of particle crushing at large strains was notably higher compared to the measurements obtained from triaxial tests. Additionally, they observed that a state of constant gradation was eventually attained at very large strains. Wei et al. [16] provided clear illustrations of the progressive evolution of particle breakage and the changes in surface smoothness within the shear band. They further demonstrated that samples of calcareous sand showed slight dilation at low shear strains and continuous contraction at high shear strains. Furthermore, the authors observed that the residual shear stress exhibited an upward trend with increasing the vertical loading stress, regardless of the material properties and shear rate.

However, the micromechanics and evolution of particle breakage in granular soils are difficult to observe at the particle level during shearing through laboratory testing. Therefore, X-ray microtomography has been applied to the crushing process at the particle scale to understand the grain-scale characteristics of calcareous sands [26,27,28]. This non-destructive approach has made significant contributions to our understanding of intraparticle contact conditions, stress redistribution, and the localization of particle breakage. 

Hall et al. [26] employed X-ray microtomography imaging to accurately quantify the initiation and progression of localized deformation processes in sand at the particle-scale resolution. Their study verified the significant role of particle rotations in strain localization. The characteristics of shear band formation and strain evolution associated with particle breakage were investigated in calcareous sands with varying levels of loading stress through the examination of scanning electron microscopy (SEM) images obtained from a series of ring shear tests.

From the above, it is clear that obtaining information about shear band evolution and micro-mechanics of particle crushing in conventional laboratory experiments is very difficult.

Recently, studies of direct shear tests at the microscopic level have become more popular using the Discrete Element Method among others by [14,29,30,31,32,33,34,35,36,37,38,39,40,41,42,43,44,45,46,47,48,49]. Seyyedan et al. [29] employed a novel approach that combined the Discrete Element Method (DEM) and Extended Finite Element Method (XFEM) to conduct numerical simulations of direct shear tests. Their study focused on 2D assemblies of angular rockfill particles capable of experiencing breakage. The findings of their study revealed that particle breakage in granular materials leads to a reduction in anisotropy, dilative behavior, and mobilized friction angle within the sample. Notably, the broken particles were observed to be distributed throughout the shear box, exhibiting a diagonal trend.

Jeong et al. [30] investigated the shear and particle crushing behavior using PFC2D (Particle Flow Code) in a ring shear apparatus. They observed that clump particles experienced partial crushing at peak shear stress, and further particle crushing occurred during shearing, leading to the production of finer particles, particularly at the residual shear stress along the shearing plane. However, this study does not show the effect of the crushing of the agglomerated particles on the evolution of the volumetric strain.

Nitka et al. [31] conducted full 3D discrete simulations using real scale cohesionless sand particles to investigate the formation of the localization zone during the initial phase of direct shear tests. Their findings revealed that, contrary to expectations, the localization zone in the pre-peak regime did not follow a straight-line pattern.

Li et al. [32] conducted comprehensive investigations using a combination of experimental and 2D discrete element method (DEM) numerical analyses to examine the shearing characteristics of a talus-like rock mass obtained from project site. Their study focused on large-scale shear tests and demonstrated that under lower normal stress and higher block content conditions, specimen dilation becomes increasingly pronounced.

These earlier studies have shown the feasibility of the DEM in evaluating the breakage effect. However, there are still some aspects that need improvement to reproduce the effect of breakage on the micro- and macroscopic behavior of crushable granular materials.

Understanding the micro-mechanics of particle breakage evolution in shearing granular soils is very useful to control the effect of breakage on shear stress and material dilatancy in both laboratory tests and engineering practices.

In this paper, a novel 3D DEM model of the direct shear test on crushable coarse sand was developed using sphere particles and crushable bonded agglomerates to demonstrate that it is possible to simulate the evolution of particle breakage in direct shear tests on crushable coarse sands with large amounts of particles, using particle flow code (PFC3D) [33]. The main purpose is to gain insights into the macro- and micro-mechanical behaviors of crushable coarse sand. To comprehensively analyze the impact of particle breakage, numerical simulations are carried out multiple times, both with and without the crushing procedure, under varying normal stresses. Under high stress, the computation results highlight the intense particle breakage and rearrangement within the shear band conditions and demonstrate that incorporating particle crushing in the model as an agglomerate is essential to replicate the realistic behavior and experimental observations of crushable sand reported in the literature.

## 2. Discrete Element Method for Particle Flow

### 2.1. Particle Contact Model

The numerical simulations in this study were performed using the commercial DEM software PFC version 7.0 [33]. This software employs the Discrete Element Method (DEM) technique, initially introduced by Cundall and Strack [34], to simulate the mechanical behavior of unbonded particles.

In DEM simulations of granular materials, the inter-particle contact law plays a crucial role in determining the macro- and micro-behavior of the entire assembly. Simulating granular materials with different contact types can be more complex than using a single contact type. In this research, three contact models were utilized for different contact types.

Considering the shape effect of sand, the rolling resistance linear contact model was adopted between sand particles, which were modeled as spheres, to reduce the computational cost and have acceptable performance. The linear parallel-bond model was applied to contacts between micro-particles within the crushable agglomerate (breakable particle). The use of crushable agglomerates can reasonably capture all crushing mechanisms that can occur. To describe the interaction between particles and walls, the linear contact model was employed. This model is a built-in constitutive model commonly used in the discrete element software PFC, which ensures compatibility with standard practices in the field.

#### 2.1.1. Linear Model

The linear model represents an interface with infinitesimal behavior that does not resist relative rotation resulting in a contact moment $M_{c}$ of zero. The force-displacement law for the linear model governs the updates the contact force and moment according to the following equations:(1)$$F_{c}=F^{l}+F^{d},M_{c}=0$$

Figure 1 illustrates the decomposition of the contact force into two components: the linear force, denoted as $F^{l}$, and the dashpot force, denoted as $F^{d}$.

The linear force can be updated using the following equation:(2)$$F_{n}^{l}=k_{n}\times ∆\delta_{n}$$
(3)$$F_{s}^{l}=k_{s}\times ∆\delta_{s}$$
where $k_{n}$ represents the normal stiffness and $k_{s}$ represents the shear stiffness. $∆\delta_{n}$ and $∆\delta_{s}$ denote the increment displacement along the normal and shear directions, respectively.

In addition, a friction coefficient μ is incorporated to enforce a Coulomb limit at the shear direction of the contact.
(4)$$F_{s}^{l}\leq \mu \times F_{n}^{l}$$

The calculation of the normal and shear contact forces between particles involves the evaluation of the dashpot model, which can be expressed as follows:(5)$$F_{n}^{d}=2\beta_{n}\sqrt{m_{c}k_{n}}{\dot{\delta }}_{n}$$
(6)$$F_{s}^{d}=2\beta_{s}\sqrt{m_{c}k_{s}}{\dot{\delta }}_{s}$$
(7)$$m_{c}=\frac{m_{1}\times m_{2}}{m_{1}+m_{2}}$$

In the equations, $m_{1}$ and $m_{2}$ denote the masses of the contact particles, while $\beta_{n}$ and $\beta_{s}$ represent the damping ratios associated with the normal and shear directions, respectively.

#### 2.1.2. Rolling Resistance Linear Model

In the DST model, a rolling resistance contact model was utilized to account for the behavior of sand particles. This model introduced rolling friction at the contacts between the simulated sand particles, as illustrated in Figure 2. By incorporating the rolling resistance contact model, the simulation could more accurately capture the behavior of coarse sand assemblies by effectively limiting relative particle rotation. Compared to the conventional linear contact model, the rolling resistance contact model provided enhanced performance in representing the behavior of the sand particles.

The rolling resistance contact model implemented in PFC3D extends the functionality of the linear model by incorporating a rolling resistance mechanism. This model introduces modifications to the force-displacement relationship to update the contact force and moment as follows:(8)$$F_{c}=F^{l}+F^{d},M_{c}=M^{r}$$

In the given equations, the linear force is denoted by $F^{l}$, the dashpot force is denoted by $F^{d}$, and the rolling resistance moment is represented by $M^{r}$. The update process for the linear and dashpot forces follows the same method as in the linear model. However, the rolling resistance moment is updated according to the following steps: initially, the rolling resistance moment is incremented by the following equation:(9)$$M^{r}∶=M^{r}-k_{r}∆\theta_{b}$$

Here, $∆\theta_{b}$ represents the relative increment in bend rotation, and $k_{r}$ represents the stiffness of the rolling resistance, which is defined as
(10)$$k_{r}=k_{s}{\bar{R}}^{2}$$

Here, $\bar{R}$ represents the effective radius of the contact, which can be defined as
(11)$$\frac{1}{\bar{R}}=\frac{1}{R^{\left(1\right)}}+\frac{1}{R^{\left(2\right)}}$$
where $R^{\left(1\right)}$ and $R^{\left(2\right)}$ represent the radii of the contact particles. In the case where one side of the contact corresponds to a wall, the radius is considered infinite $R^{\left(2\right)}=\infty$. Subsequently, after updating the rolling resistance moment, its magnitude is compared to a predefined threshold limit to ensure its validity:(12)$$M^{r}=\left\{\begin{array}{c} M^{r},\;∥M^{r}∥\leq M^{*}\\ M^{*}(M^{r}/∥M^{r}∥),\;otherwise \end{array}\right.$$

The limiting torque is defined as follows:(13)$$M^{*}=\mu_{r}\bar{R}F_{n}^{l}$$

Here, the coefficient of rolling resistance is $\mu_{r}$, while the normal linear force is $F_{n}^{l}$.

#### 2.1.3. Linear Parallel-Bond Model

The linear parallel bond operates alongside the linear contact model, except when the maximum stresses exceed the bond strength. In such cases, the bonding is canceled, and only the linear contact model remains between the contact particles. A visual illustration of this behavior is presented in Figure 3.

The contact force is divided into three components: the linear force $F^{l}$, the dashpot force $F^{d}$, and the parallel-bond force. Similarly, the contact moment $M_{c}$ is represented by the parallel-bond moment M. The updates for the linear and dashpot forces follow the same approach as in the linear model. The parallel bond force $\bar{F}$ is further resolved into the normal force ${\bar{F}}_{n}$ and the shear force ${\bar{F}}_{s}$, while the parallel bond moment $\bar{M}$ is resolved into the bending moment ${\bar{M}}_{b}$ and torsional moment ${\bar{M}}_{t}$:(14)$$\bar{F}=-{\bar{F}}_{n}{\hat{n}}_{c}+{\bar{F}}_{s}$$
(15)$$\bar{M}=-{\bar{M}}_{t}{\hat{n}}_{c}+{\bar{M}}_{b}$$

The following sections outline the processes involved in determining particle motion, contact and tensile forces, and rotational moments.

The normal force is updated: (16)$${\bar{F}}_{n}={\left({\bar{F}}_{n}\right)}_{0}+{\bar{k}}_{n}\bar{A}∆\delta_{n}$$

Here, ${\left({\bar{F}}_{n}\right)}_{0}$ represents the initial parallel component of the normal force at the start of the time step, $\bar{A}$ denotes the bond area, ${\bar{k}}_{n}$ represents the normal stiffness, and $∆\delta_{n}$ represents the increment in normal displacement. The shear force undergoes an update through the following process:(17)$${\bar{F}}_{s}={\left({\bar{F}}_{s}\right)}_{0}-{\bar{k}}_{s}\bar{A}∆\delta_{s}$$

Here, ${\left({\bar{F}}_{s}\right)}_{0}$ represents the parallel component of the initial parallel shear force component at the start of the time step, ${\bar{k}}_{s}$ represents the shear stiffness, and $∆\delta_{s}$ represents the increment in shear displacement. The update for the torsional moment is given as follows:(18)$${\bar{M}}_{t}={\left({\bar{M}}_{t}\right)}_{0}-{\bar{k}}_{s}\bar{J}∆\theta_{t}$$

Here, ${\left({\bar{M}}_{t}\right)}_{0}$ represents the initial twisting moment at the start of the time step, $\bar{J}$ denotes the polar moment of inertia of the bond, and $∆\theta_{t}$ represents the relative twist rotation. The update for the bending moment is given as follows:(19)$${\bar{M}}_{b}={\left({\bar{M}}_{b}\right)}_{0}-{\bar{k}}_{n}\bar{I}∆\theta_{b}$$

Here, ${\left({\bar{M}}_{b}\right)}_{0}$ represents the initial bending moment at the start of the time step, $\bar{I}$ denotes the moment of inertia of the bond, and $∆\theta_{b}$ represents the incremental bend-rotation.

## 3. DEM Simulations of Direct Shear Test

In this study, the developed Discrete Element Method (DEM) model for the direct shear test (DST) was used to investigate the effect of particle breakage on the shear strength and dilatation of the granular material under different normal high stresses of 160, 200, 300, 400, 600, 800, 1200 and 1600 kPa.

In order to establish a numerical model and determine the micro-parameters of the material, 3D DEM simulations were conducted using the Particle Flow Code (PFC3D) software by Itasca [33] to simulate the results of direct shear tests in this study. The experimental data from direct shear tests conducted by Salazar et al. [35] were chosen for calibration and validation of the numerical DEM model. This process involved establishing the correlation between microscopic and macroscopic parameters utilized in the model.

### 3.1. Model Setup

The DEM model setup of a $60\times 60\times 24\;{mm}^{3}$ direct shear box in this study is shown in Figure 4. The model is composed of two distinct parts, namely the upper box and the lower box. The top wall of the upper box is designed to move freely up and down, and its role is to maintain the normal stress through a servo-controlled system during compression and shearing, as performed in conventional experimental tests. The upper box remains stationary, while the lower box performs shear displacement by moving horizontally from left to right at a constant velocity. To ensure the accuracy of the simulation, two additional walls were installed on either side of the shear box to prevent the balls from falling out into the gap created during shearing. To avoid any unintended effects on the simulation, the friction coefficient of the wall constituting the shear box was set to 0.

The mechanical behavior of granular materials is significantly influenced by the shape of their particles. However, using non-spherical particles in simulations can be computationally intensive and require significant storage capacity [36,37]. Nevertheless, recent research has demonstrated that using sphere particles with appropriate rolling friction can lead to similar effects as non-spherical particles while reducing both storage requirements and computational time [21,37,38]. Therefore, to accurately capture the localization of a large number of particles and minimize computational costs, the granular material is modeled using assemblies of spheres.

In this study, the grain size of particles is often up-scaled due to computational costs and previous research suggesting that up-scaling to a reasonable factor has a minimal impact on experimental results [35,39,40,41,42]. Given the focus on investigating the effect of grain crushing and the size of the model, the diameter of each sphere was scaled up by a factor of two and selected based on the grain size distribution of the tested sand. To ensure consistency across all numerical analysis models conducted in this study, the same number of particles was applied in all simulations.

The numerical simulation of the direct shear test involved three main steps: specimen preparation, consolidation, and shearing. In the specimen preparation phase, the dimensions of the box were defined, and spheres were randomly distributed inside the box using the PFC porosity command to achieve the target porosity regardless of overlap. Subsequently, the overlap between particles was gradually reduced as particles were allowed to move and rearrange, ensuring an optimal fit within the boundaries of the simulation box. Furthermore, in order to minimize the inter-particle strength during the particle generation process, the friction between particles was set to 0 to allow particles to rearrange until. Then, the particles fell under the action of gravity force. A local damping coefficient was implemented in the simulation to effectively the system’s kinetic energy. Finally, several calculations were needed until an equilibrium state was reached. After that, the contact model parameters were fixed to their final values. The rolling resistance linear contact model included in PFC3D was utilized to define the local contact between individual particles to account for the particle roughness effect.

The consolidation process was performed under a predefined normal stress applied using a servo-controlling loading mechanism. This mechanism adjusts the speed of the top wall to achieve the desired reaction force in the vertical direction corresponding to the targeted normal stress levels (160, 200, 300, 400, 600, 800, 1200, and 1600 kPa). After consolidation, the shearing process was performed with a constant shear rate of 0.01 m/s to attain a horizontal displacement of 10 mm. The loading rate was intentionally selected to be sufficiently slow to ensure that the test was conducted under quasi-static conditions. Horizontal shear stress is recorded automatically during the numerical simulation. Horizontal displacement is determined by tracking the displacement in the x-direction of the lower box, while vertical displacement is measured by monitoring the displacement in the z-direction of the top wall.

The normal stress $\sigma_{n}$ and shear stress $\sigma_{s}$ acting on the shear plane can be determined through the following equations:(20)$$\sigma_{n}=\frac{F_{n}}{B(L-\vartheta t)}$$
(21)$$\sigma_{s}=\frac{F_{s}}{B(L-\vartheta t)}$$

Here, $F_{n}$ represents the normal force applied to the shear plane, corresponding to the imposed normal load on the sample. $F_{s}$ denotes the shear force, which is equivalent to the horizontal forces acting on the upper box. L and B refer to the length and width of the shear box, respectively. ϑ represents the constant displacement rate of the lower box, and t signifies the duration of the test.

### 3.2. Calibration and Validation Results

In PFC3D, the behavior of the granular material at the macroscopic level is derived from the interactions of its microscopic properties. To obtain accurate results, it is important to calibrate the micro-scale parameters used in the contact particle model of the PFC3D simulation. In previous work of Benmebarek and Movahedi [42], the coarse sand was calibrated using the direct shear test simulation and validated against Salazar’s experimental work for different confinement pressures of 40, 80 and 160 kPa. The rolling resistance linear contact model was chosen for the DEM simulations, and the trial-and-error method was utilized to adjust the main input microscopic parameters until the desired macroscopic behavior was accurately replicated.

In the current study, the calibration process was conducted to facilitate the implementation of crushable particles, aiming to streamline its usage. For the sake of simplicity, the model focused on five specific diameters, ranging from 2 to 4 mm, to ensure a uniform particle-size distribution. Subsequently, numerical simulations were performed on the specimen, and the obtained results were compared with experimental data derived from Salazar’s study, specifically under a normal stress condition of 160 kPa. Table 1 presents the calibrated parameters of the DEM model.

Figure 5a illustrates that the model successfully aligns with the experimental shear stress path throughout the entire simulation. Both the initial porosity and the stiffness of the contact particles can affect the initial slope at low strain according to calibration tests. On the other hand, the shear strength can be influenced by adjusting the coefficients of particle friction and rolling resistance [42]. Figure 5b depicts the vertical displacement, which signifies the overall volume change and represents the phenomenon of contractive dilatancy, through the shearing process. The shape of the curve closely resembles the response typically observed in dense sand, characterized by initial contractive behavior at low strain followed by a transition to dilatant behavior.

## 4. Particle Breakage on the Shear Band

Obtaining information about the micro-mechanics of particle breakage through laboratory experiments is a challenging task that requires a significant amount of effort. However, the DEM provides an alternative solution for simulating particle crushing. The location of the horizontal shear band during DST is still a topic of research, and it is suggested that it is aligned along the vorticity axis.

Based on previous research [43,44,45,46], shear localization is often assumed to occur along a straight line enforced by the boxes, resulting in particle breakage typically occurring at the shear plane where higher shear forces are located. To simulate the phenomenon of crushing on the shear band, crushable agglomerates were introduced in place of the particles located on the potential horizontal shear band. In this study, various crushable agglomerates with the same diameter as the original particles were formed using a large number of micro-grains (approximately 50 micro-grains of the same size) that are bonded together in accordance with the crushable parallel bond properties, allowing for the consideration of all possible crushing mechanisms. The diameter and number of micro-grains were varied to match the diameter of the agglomerate being simulated. In the DEM simulation, the crushing of granular material was defined as breakage if one or more particles were separated from the crushable agglomerate.

### 4.1. DST Tests with Various Normal Stresses

In this study, direct shear test simulations were conducted on crushable coarse sand under different normal stresses from medium to high (160, 200, 300, 400, 600, 800, 1200, and 1600 kPa). The aim was to investigate the effect of particle breakage on the shear strength, dilatation of the granular material, and evolution of grain crushing through shearing. To achieve this, the changes in these properties were compared between simulations where particle crushing was considered and those where crushing was not. The parameters of the micro-grains were chosen based on experience, given the qualitative nature of the study. To ensure the integrity of the crushable model, a calibration was conducted, wherein the agglomerate was designed not to break under normal stress levels of 160 kPa or below. However, when subjected to normal stress exceeding 160 kPa, the particles will undergo a gradual breakage process in accordance with the applied stress level. The parameters of the micro-particles forming the crushable agglomerate are shown in Table 2.

### 4.2. Single Grain Crushing Test

The simulation involved individual particles and plates for focusing on highly stressed particles (Figure 6). For modeling the loading plates in diametral compression, rigid plate walls were used. During the loading phase, the top plate wall experienced a constant velocity of v=0.01 m/s, while the bottom plate wall remained fixed. In Figure 6a, it can be observed that the applied force on the agglomerate increases with vertical displacement until it reaches a peak value, after which it abruptly drops. Random rotations and testing of the sample and similar results were obtained, confirming the absence of weak zones within the agglomerate. Figure 6b shows that the agglomerate splits into two major parts upon reaching the peak force. These findings align with the experimental results obtained from the diametral compression test, as reported by Kundu et al. [47] and Liu et al. [14].

## 5. Results and Discussion

### 5.1. Shear Stress

The shear stress versus lateral displacement curves displayed in Figure 7 indicate that the simulations reproduce the well-known behavior of sand in direct shearing when subjected to an increase in normal stress. The plot indicates that the peak shear strength was achieved at a small horizontal displacement and increased with the normal stress, particularly for the non-crushable model (Figure 7a). In contrast, the curves of the crushing particle model displayed a more gradual shape primarily attributed to particle crushing (Figure 7b), as also observed in the numerical study by Jo et al. [48].

Interestingly, at normal stresses of 160 and 200 kPa, the difference between shear stress curves for both the crushable and non-crushable models were negligibly small, suggesting minimal or no occurrence of particle crushing within the crushable model. However, for vertical stresses of 300 kPa or higher, a significant reduction in shear stress was observed in the crushable agglomerate model compared to the model where crushing was not considered. This phenomenon highlights the intense particle breakage and rearrangement within the shear band under high-stress conditions. It can be concluded from Figure 7 that the degree of particle crushing increases with the normal stress [23,49]. The decrease in stress can be attributed to reduced interlocking between particles due to particle breakage. Even when subjected to equal vertical stress, the assembly with crushing particles is expected to exhibit lower shear stress compared to the non-crushable sample. The difference in shear stress between the two samples becomes more pronounced at higher stresses, indicating greater particle crushing within the shear band. These macroscopic results are qualitatively consistent with observations from conventional experimental direct shear tests.

### 5.2. Volumetric Strain

Figure 8 presents vertical displacement versus lateral displacement curves for various normal stresses. The non-crushable simulation curves depicted in Figure 8a exhibit minimal variations among themselves, with shearing at all normal stress levels leading to dilation. This behavior arises from the particles’ inability to break, coupled with the fact that the samples assume nearly identical dense packing when the confining pressure is applied.

From Figure 8b, the crushable model exhibited a slight decrease in sample dilatation at the beginning of tests at normal stresses of 160 and 200 kPa due to specimen contraction. Subsequently, the assembly displayed pronounced volumetric dilation during the lateral displacement phase, this effect of volume dilation was particularly pronounced in tests with low-stress levels. However, at higher stress levels of 300 kPa or more, a contractive behavior dominated the specimen deformation, resulting in large volumetric strains at high shear strains, which is consistent with the characteristic of particle breakage within the shear band observations of Coop et al. [4], Tarantino and Hyde [23], and Qin et al. [24]. For tests at 800 kPa normal stress or more, the contractive behavior slowed down, showing gradually stabilized volumetric deformation. This trend is consistent with the observations by Coop et al. [15] and Wei et al. [16,20] that tests at higher stress levels can reach stable volumetric strains at relatively lower shear strains.

The reduction of dilative behavior is related to particle breakage occurrence. Increasing the vertical load leads to an increase in particle breakage, and the creation of smaller particles due to breakage resulted in the reduced volume and porosity of the assembly as the gaps between primary particles were filled rather than from the occurrence of expansion behavior due to interlocking. These observations align with similar experimental findings reported by Qin et al. [24]. In contrast, simulations where particle crushing was not considered showed a consistent pattern of slight initial sample contraction followed by dilative behavior throughout the shearing process, regardless of the applied normal stress. This behavior indicated an increase in specimen porosity due to interlocking between particles.

The findings highlight the significant influence of particle crushing on the macroscopic behavior of soil, particularly in terms of strength and strain. This impact is particularly prominent in direct shear tests, where particle breakage reduces the rate of dilation.

### 5.3. Visualization of Particle Crushing Evolution

After conducting the DST simulations, the deformed specimens were visualized from the virtual DST box. Simulation results were obtained from both the lateral and inner views of the shear zone band in the middle of the specimen, where the crushable particles were located. Visualizing the micro particles provides a more comprehensive understanding of the macroscopic behavior of material crushing.

Figure 9 displays the lateral and inner views of the samples, highlighting the broken agglomerates. These figures reveal the increasing degree of crushing experienced at different normal stresses. They were captured at a shearing displacement of 10 mm, with all crushed agglomerates highlighted within the shear band zone.

Based on Figure 9a, no crushing was observed for the vertical stress of 160 kPa, and only a few cracks were found inside the agglomerates. However, these cracks did not significantly affect the shear stress and dilatation, and the results were similar to those obtained when crushing was not considered. This suggests that cracks inside a particle at low pressure reduce the element’s stiffness, but as long as the particle remains unbroken, they have no significant impact on the particle assembly. At 200 kPa, some crushing started to appear at the edges of the DST, slightly affecting the shear stress and dilatation (Figure 9b).

For normal stresses of 300 and 600kPa (Figure 9c–e), a considerable amount of particle crushing occurred throughout the shear band zone, generating smaller particles with different sizes than the primary particles. These smaller particles filled the empty spaces between particles, reducing the sample volume. Consequently, the dilative behavior of the sample decreased, and a contractive behavior was observed at high pressures due to particle breakage. At a pressure of 800kPa or higher (Figure 9f–h), most of the particles were completely crushed, resulting in the production of a large amount of fine particles and a more densely packed sample. The phenomenon of particle crushing near the shear plane has been verified in previous studies on the DST of granular materials.

Additionally, the figure shows that large crushed particles are concentrated in the center of the shear box, while small particles migrate up and down through the empty spaces and accumulate mostly at the lower part of the box due to the gravity effect during shearing. These numerical observations confirm the results of experimental shear tests [25,50,51].

The observations made in Figure 10 revealed that particle breakage became more pronounced with increasing shear displacement, indicating the influence of shearing on the crushing behavior. Additionally, the results confirmed our initial expectations, showing an increase in crushing with increasing vertical stress. However, it is noteworthy that the rate of increase in crushing gradually decreased as the normal stress increased, which aligns with the findings reported by Tarantino and Hyde [23] in their experimental study. These consistent trends between the simulation results and the previous experimental observations provide further validation of the crushing behavior under varying loading conditions.

## 6. Conclusions

In this study, a 3D DEM of DST on breakage granular material was performed using the PFC3D code to explore the effect of particle breakage evolution under different normal stresses on the shear strain and volume change during shearing. The implementation of crushable particles represented by agglomerates in the horizontal shear band of the DST was found to be crucial in accurately capturing the material’s response under high normal stresses. Based on the 3D DEM computation results, the following main conclusions may be drawn:Particle crushing is essential to replicate the realistic behavior of sand, especially volumetric contraction, in high normal stress shear tests;With an increasing normal stress, the difference in the shear strength and volumetric strain of the crushable model and the non-crushable model increases;Crushable sand behaves similarly to non-crushable sand at low normal stresses, as long as the stress is not sufficient to cause particle crushing;Increasing the normal stress in DST causes a transition in the soil’s volumetric strain during shearing from dilative to contractive behavior for crushable sands;Particle breakage inside DST reduces shear strength and dilatancy, resulting in a less stiff shear stress–strain response;Shear stress increases with an increase in normal stress, regardless of whether crushable or non-crushable models are used;Particles on the shear band are partially crushed at peak shear stress, and further shearing produces finer particles along the shearing plane;3D DEM simulations visually demonstrate the generation of smaller particles due to grain crushing, which fill pore spaces, thus reducing assembly dilatancy, and the effect of particle interlocking. At higher normal stress levels, the volumetric strain converges to a stable value;DEM simulations of DST which include crushable particles align well with laboratory experiments and provide physically realistic macro-scale results.

## Figures and Tables

**Figure 1 materials-16-05025-f001:**
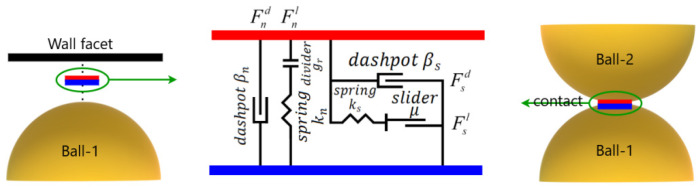
Illustration of the linear contact model and locations of contact planes for two fundamental contact types.

**Figure 2 materials-16-05025-f002:**
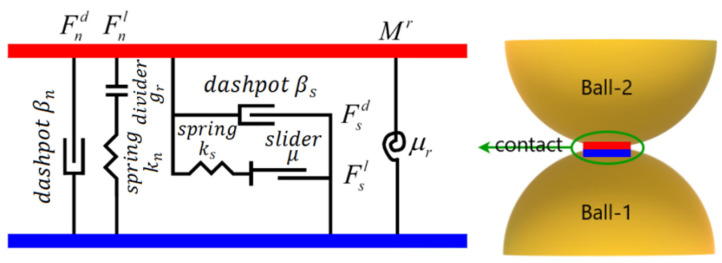
Illustration of the rolling resistance linear model.

**Figure 3 materials-16-05025-f003:**
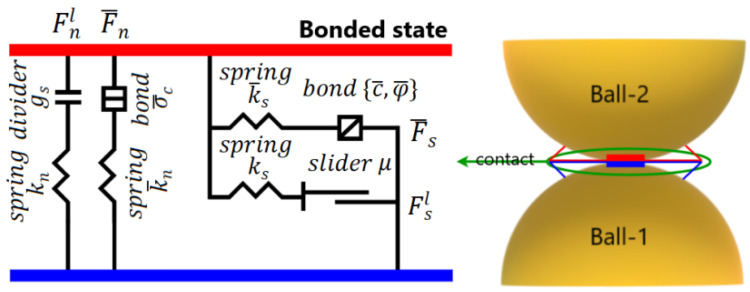
Illustration of the linear parallel contact model.

**Figure 4 materials-16-05025-f004:**
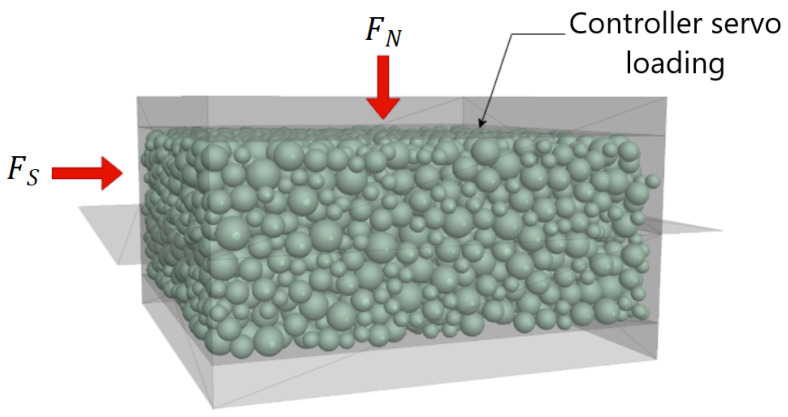
Virtual DEM model of Direct Shear Test.

**Figure 5 materials-16-05025-f005:**
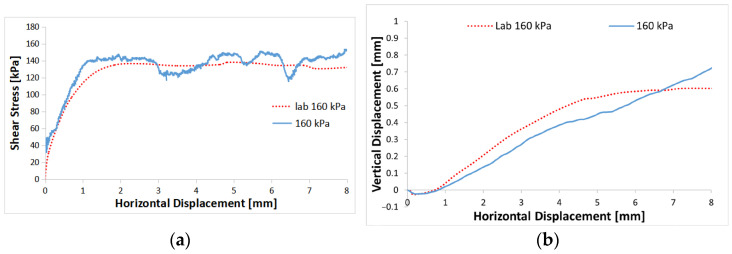
Comparisons between the DEM simulation and experiment (Salazar et al. [34]) for normal stresses of 160 kPa: (**a**) relationship between shear stress and shear strain; (**b**) relationship between volumetric strain and shear strain.

**Figure 6 materials-16-05025-f006:**
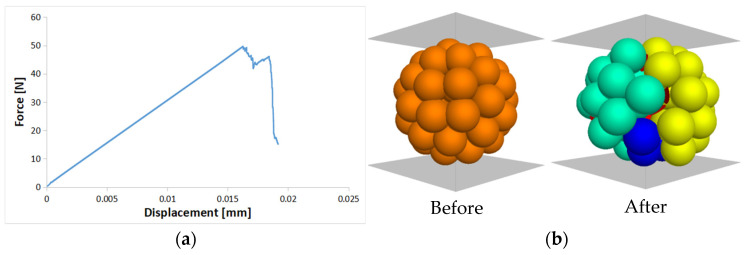
(**a**) Force-displacement curve of single agglomerate; (**b**) fracture pattern of the macro-grain at the peak load.

**Figure 7 materials-16-05025-f007:**
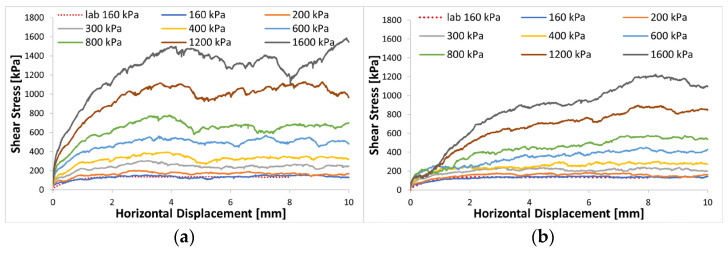
Shear stress curves of different vertical stress: (**a**) non-crushable; (**b**) crushable.

**Figure 8 materials-16-05025-f008:**
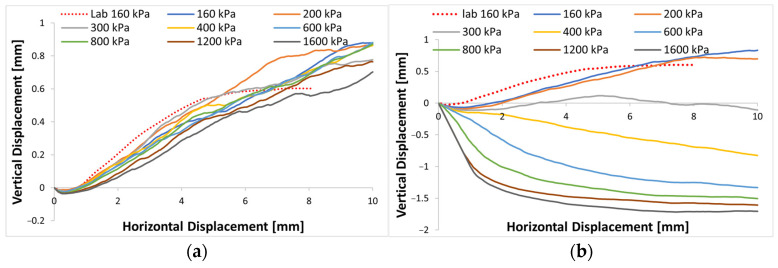
Dilatation curves of different vertical stress: (**a**) non-crushable; (**b**) crushable.

**Figure 9 materials-16-05025-f009:**
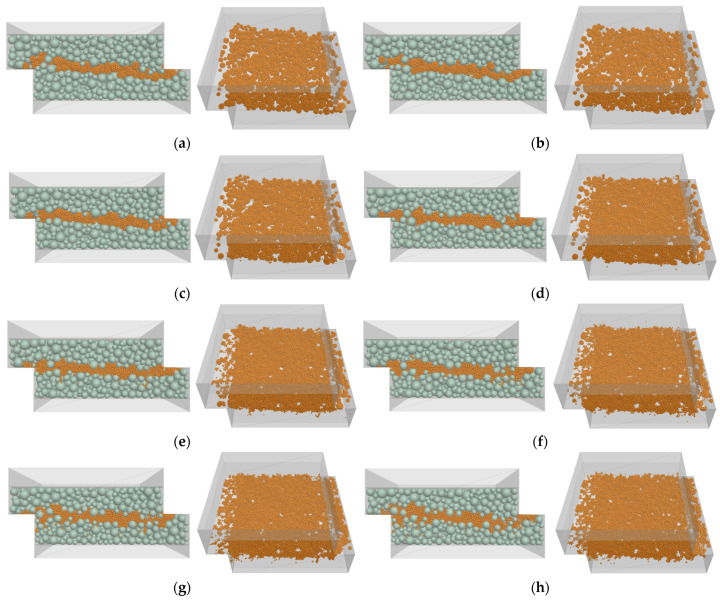
Visualization of crushing evolution on the shear band of DST: (**a**) Normal stress = 160 kPa, ratio of broken bonds = 0.04. (**b**) Normal stress = 200 kPa, ratio of broken bonds = 0.09. (**c**) Normal stress = 300 kPa, ratio of broken bonds = 0.31. (**d**) Normal stress = 400 kPa, ratio of broken bonds = 0.53. (**e**) Normal stress = 600 kPa, ratio of broken bonds = 0.69. (**f**) Normal stress = 800 kPa, ratio of broken bonds = 0.73. (**g**) Normal stress = 1200 kPa, ratio of broken bonds = 0.76. (**h**) Normal stress = 1600 kPa, ratio of broken bonds = 0.79.

**Figure 10 materials-16-05025-f010:**
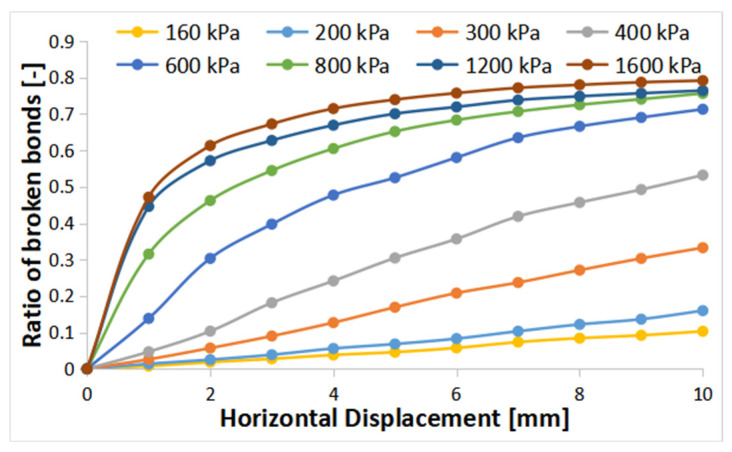
The evolution of broken bonds during shearing.

**Table 1 materials-16-05025-t001:** The microscopic parameters for DEM simulation.

Particles Properties		
Elementary particles size, D	mm	2.4, 2.8, 3.2, 3.6, 4
Effective modulus of sand particle, E*	GPa	2
Normal-to-shear stiffness ratio of sand particle ${K_{n}/K}_{s}$	-	1
Density	$kg/m^{3}$	2600
Damping coefficient	-	0.5
Friction coefficient between sand particles, μ	-	0.3
Rolling resistance coefficient of sand particle, $\mu_{r}$	-	0.25
Porosity	-	0.41
Effective modulus of wall particle, E*	GPa	4
Normal-to-shear stiffness ratio between wall particle ${K_{n}/K}_{s}$	-	1
Friction coefficient between sand particle and wall, μ	-	0.0

**Table 2 materials-16-05025-t002:** Parameters of micro-particles.

Particles Properties		
Elementary particle size, D	mm	0.46, 0.58, 0.64, 0.74, 0.84
Young’s modulus of the particle, E*	GPa	2
Ratio of normal to shear stiffness of the particle, ${\bar{K}}_{n}$/${\bar{K}}_{s}$	-	1
Density	$kg/m^{3}$	2650
Damping coefficient	-	0.5
Particle friction angle coefficient, μ	-	0.5
Particle Friction coefficient, μ between particle and wall	-	0
Parallel bond properties		
Installation gap, g¯i	m	0
Young’s modulus of the parallel bond, $\bar{E}$	GPa	2
Ratio of normal to shear stiffness of the parallel bond, ${\bar{K}}_{n}$/${\bar{K}}_{s}$	-	1
Friction angle of the parallel bond	°	20
Tensile bond strength	MPa	5
Cohesion bond strength	MPa	5

## Data Availability

The datasets, which were generated during and analyzed in the current study, are available in the main manuscript, and any additional details can be obtained from the authors.

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
