# Peer review of "3D DEM Analysis of Particle Breakage Effect on Direct Shear Tests of Coarse Sand"

_materials, 2023, doi:10.3390/ma16145025_

Round 1
Reviewer 1 Report
Reviewers' comments:
This manuscript presents some numerical simulations of the particle breakage process in medium and high stressed direct shear tests of coarse sands by the crushable bonded agglomerates. The results highlight the formation of the shear band and the effect of breakage on the micro- and macro-behavior under various levels of normal stress. A good consistency can be observed with experiment results in literatures. Thus, the reviewer believes that this manuscript can be published in the Materials after revised. Here are some more detailed comments that might be considered for the revision.
(1) In introduction section, several paragraphs are used to review relevant research progress. These paragraphs with similar topic should be merged into one paragraph. In addition, the creation of this works is insufficient, therefore, the author needs to emphasize the innovation of this work in the introduction section.
(2) The three contact models mentioned in the manuscript were not first proposed by the author, so please add references, including theoretical formulas.
(3) There are too many conclusion points, some of which can be merged. The conclusion section needs to be rewritten.
(4) The figure 9 should be recombined. The distribution of subgraphs on several pages can affect reading. In addition, can fragmented particles be displayed in different colors?
(5) It is recommended that the author calculate the particle size distribution of broken fragments as a function of different normal pressures.
(6) Other small suggestions:
ž Line 263: “60?60?24 ??” should be “60×60×24 ??”.
ž Part of the table 1 is missing, please improve the table.
ž Please check the shear rate of 0.01 m/s, it may be too large. Or why is it set to 0.01 m/s, referring to the experiment?
Reviewer 2 Report
In the research paper, “3D DEM Analysis of Particle Breakage Effect on Direct Shear Tests of Coarse Sand” the authors attempted to model the shearing of the sand particles based on DEM modelling of spherical particles.
The article is mostly well written and easy to follow. However, I have some doubts about the physical validity of the simulations performed. In general, the outcome does not seem to be much affected in the shearing process, which was confirmed by the comparison between the DEM and the experimental Fig. 5. Below are some of my the remarks I have regarding this article.
Line 287 – As far as I know, increasing the size of the particle is not as straight forward as the Authors make it out to be, because it affects other particle parameters as well. Therefore, scaling is allowed, but requires a careful approach, and recalibration of the other material parameters to keep the physics of the process intact.
Line 296 – The process of generating the initial bed should be explained in more detail. I assume that the Authors' approach was to obtain the porosity of the sand, with the spherical polydisperse particles.
To what extent does this approach affect on the results obtained and on the reality of the process? It is known that the density of the material (and the porosity) has a direct effect on the shear strength, which the authors only confirmed in their results by changing the compaction of the material.
At what point was the particle-particle friction and force calculation (overlap) was re-enabled?
Line 297 – As the Authors used a linear contact model, there was no other choice but to save the numerical stability and use the global dumping factor. Was it also left on for the compression and shear phases?
Line 305 – With or without friction?
Line 307 – Eurocode suggests 0.04 mm/s, but due to the limitations of the DEM and the computational time required, the 10 mm/s, given by the authors, should still be acceptable.
Lines 361, 362 – Could the Authors elaborate a bit on how they created bonded particles, i.e. position of the spheres within the particle, contact overlap, bond strength? Fig. 9 Was the bonded particle created once, and regenerated in multiple instances, or was there some randomness involved?
Some of the conclusions are more a statements and well known facts rather than the real conclusions proven by the analysis carried out.
In my opinion, there is some potential in the research presented, but the analyses are not yet complete. Having such a powerful tool as DEM at hand, the Authors might consider perform a more detailed analysis of the simulations. For example, the time evolution of the stress tensor distribution within the particle bed during particle fracture could provide some interesting insights into the process presented.
Reviewer 3 Report
The authors simulated direct shear tests using 3D Discrete Element Method (DEM). A comparison between non-crushable and crushable particle models was made. The authors should consider the following points:
1) The following sentence could be revised, the meaning isn't clear: "the micro-mechanics of particle breakage evolution in shearing granular soils are necessary to understand the effect of breakage on shear stress and material dilatancy in both laboratory tests and engineering practices" → Perhaps it should be, "it is necessary to understand the micro-mechanics of particle breakage ...".
2) The abbreviation PFC3D was first mentioned in the last paragraph of Introduction and no explanation was given about what it stands for. This is the name of the software used in the simulations which should be made clear earlier in the text.
3) The significance of the research should be emphasized in the last paragraph of the Introduction.
4) In Section 2.1. "macro- and - behavior of the entire assembly" → Should be "macro- and micro-behavior of the entire assembly".
5) There is a typo in Eq. 5 in the notation of the normal stiffness.
6) Lines 283 to 285 seem out of place and shouldn't be a separate paragraph.
7) Line 306: "a constant shear rate of 0.01 ?/??? to achieve a horizontal displacement of 10 ??" → What was the total duration of the simulation?
8) Unnecessary hyphen symbols such as in "micro-scopic" should be removed.
9) What are the dimensions of the domain pictured in Figure 4?
10) Vertical lines should be removed from table borders.
The English language is satisfactory.
Round 2
Reviewer 1 Report
There are still some typos that need to be corrected. However, the quality of the revised paper has been improved and can be published in Materials.
Reviewer 2 Report
The authors clarified the most problematic and questionable parts of the manuscript, therefore it could be accepted for publication.
Author Response
We sincerely thank the editor and reviewer again for reviewing our manuscript and providing constructive feedback to improve our manuscript.